# Thermoregulation Effects of *Phoneutria nigriventer* Isolated Toxins in Rats

**DOI:** 10.3390/toxins16090398

**Published:** 2024-09-18

**Authors:** Carla Bogri Butkeraitis, Monica Viviana Abreu Falla, Ivo Lebrun

**Affiliations:** Laboratory of Biochemistry and Biophysics, Butantan Institute, Avenida Vital Brazil 1500, Butantã, São Paulo 05503-900, SP, Brazilmonica.falla.esib@esib.butantan.gov.br (M.V.A.F.)

**Keywords:** spider, *Phoneutria nigriventer*, toxins, thermoregulation, central nervous system, rat

## Abstract

Body temperature is primarily regulated by the hypothalamus, ensuring proper metabolic function. Envenomation by *Phoneutria nigriventer* can cause symptoms such as hypothermia, hyperthermia, sweating, and shivering, all related to thermoregulation. This study aims to analyze and identify components of the venom that affect thermoregulation and to evaluate possible mechanisms. Rats were used for thermoregulation analysis, venom fractionation by gel filtration and reverse-phase chromatography (C18), and sequencing by Edman degradation. The venom exhibited hypothermic effects in rats, while its fractions demonstrated both hypothermic (pool II) and hyperthermic (pool III) effects. Further separations of the pools with C18 identified specific peaks responsible for these effects. However, as the peaks were further purified, their effects became less significant. Tests on U87 human glioblastoma cells showed no toxicity. Sequencing of the most active peaks revealed masses similar to those of the Tachykinin and Ctenotoxin families, both known to act on the nervous system. The study concludes that molecules derived from venom can act synergistically or antagonistically. Additionally, toxins that affect thermoregulation are poorly studied and require further characterization. These toxins could potentially serve as sources for the development of new thermoregulatory drugs.

## 1. Introduction

Animal toxins are molecules selected from thousands of years of evolution. Some of their properties are quite impressive when considering some aspects, such as the amount of venom injected into the prey or victim, and the specificity of their effects, including hemostasis, cardiovascular system, and central nervous system. Some of the molecules and effects have been used as templates for the development of new drugs, such as Captopril and Ziconotide, for example. The effects on the central nervous system are more impressive if we consider that the molecules must be resistant to plasmatic enzymes and need to cross the blood–brain barrier. Some venom of snakes, scorpions, and spiders exhibit the capability to elicit potent effects on the CNS, including the death of the prey/victim. Some of the effects of the armed spider *Phoneutria nigriventer* relating to the peripheral and central nervous system have been described, but some effects clinically described in accidents were lacking, in particular when we refer to changes in the thermic regulation of the body observed in mammals and also in some clinical cases in humans [1]. Some toxins from *Phoneutria nigriventer* showed effects related to calcium and ion channels, revealing effects such as flaccid paralysis, penile erection, and hypersalivation [1,2], but no studies on body temperature have been performed even in clinical reports where variations in body temperature were described in the literature. This phenomenon and its mechanism remain totally unclear; instead, its relevance to homeostasis and physiological functions has been discussed. The main thermal effects and alterations were related to inflammatory mediators and, in some cases, could be related to specific diseases such as malignant hypertension or cerebral tumors [3]. Envenoming with *Phoneutria nigriventer* displays less significant inflammatory symptoms, suggesting another putative pathway to the body temperature alterations. Clinically, there is much evidence that *Phoneutria nigriventer* toxins, besides other observed effects, could act in thermoregulation parameters.

There are several studies characterizing the fractions of *Phoneutria nigriventer* venom. *Phoneutria nigriventer* venom has three main described neurotoxic fractions (PhTx1, PhTx2, and PhTx3), with PhTx1 and PhTx2 causing tail elevation in experimental animals, agitation, and spastic paralysis of the posterior train. PhTx2 also causes salivation, tearing, priapism, convulsions, and spastic paralysis of the anterior train, and PhTx3 causes only flaccid paralysis of the limbs [4,5].

Regarding clinical aspects, Antunes and Malaque [5] described phoneutrism as presenting predominantly local symptoms, with a systemic condition occurring mainly in children. In general, symptoms vary in degree and are classified as mild, moderate, and severe, with manifestations ranging from pain, edema, erythema, and sweating in the area of the bite to profuse vomiting, priapism, diarrhea, bradycardia, hypotension, cardiac arrhythmia, acute pulmonary edema, and shock. In the systemic context, among other symptoms, generalized sweating is observed in accidents of moderate to severe intensity. In a survey carried out between 1989 and 1998, systemic changes with the presence of sweating occurred in 7.5% of cases, with a prevalence in young people over 15 years of age [4].

Therefore, *Phoneutria* venom produces hemodynamic changes of central and peripheral origin. The central component appears to be mediated by the activation of cardiovascular centers, causing increased sympathetic discharge in the periphery, while the peripheral component appears to result from the direct activation of vascular adrenoceptors and/or release of catecholamines from sympathetic nerve endings [5].

In this work, we tested the effects of crude and purified *Phoneutria nigriventer* venom fractions on the body temperature and tested if the effects are related to inflammatory pathway or not, using regular clinical anti-thermal compounds.

## 2. Results

### 2.1. Evaluation of the Activity of Crude Phoneutria nigriventer Venom on the Rat Body Temperature Variation

The crude venom of *Phoneutria nigriventer* was subjected to in vivo tests to evaluate body temperature variation in rats. In this way, it was possible to observe that the crude venom of *Phoneutria nigriventer* caused a significant decrease in the body temperature variation of the rats, especially between 105 and 180 min (Figure 1).

### 2.2. Venom Fractionation and Assessment of Pool Activity on the Rat Body Temperature Variation

The *Phoneutria nigriventer* venom was fractionated by gel filtration, which presented five peaks (Figure 2), which were collected and analyzed by SDS-PAGE electrophoresis (Appendix A). Subsequently, the collected peaks were subjected to evaluation of body temperature variation in rats (Figure 2). Peaks II and III showed significant differences in relation to the control. Peak II showed significant hypothermic activity (*p* < 0.05), as evidenced by the average body temperature at 105 min and between 135 and 180 min. In contrast, pool III caused a significant increase in the animals’ body temperature (*p* < 0.05) after 135 min of testing.

### 2.3. Evaluation of the Rat Body Temperature of the HPLC Isolated Peaks Obtained from Pool II and III from the Venom of Phoneutria nigriventer

Peaks II and III of *Phoneutria nigriventer* venom, which showed thermal activity in rats, were separated by HPLC (Appendix A). The chromatography of peak II showed approximately 32 peaks, while the fractionation of peak III showed 33 peaks in total. The peaks were collected to analyze the rats’ body temperature variation.

The peaks collected from pool II of *Phoneutria nigriventer* showed variations in the temperature of the treated animals, as shown in Figure 3. Significant hyperthermic activity was observed in the animals treated with peaks 8 (150 and 180 min) and 13 (150 min).

Peak 28/29 from pool III of *Phoneutria nigriventer* showed a tendency for the animals’ temperature to decrease, while subpeaks 30 and 31 showed hyperthermic activity in the rats (Figure 4). Peak 30 showed greater activity at 120 min of the experiment, also standing out at the point corresponding to 150 min, while peak 31 showed activity mainly after 120 min and persisted until 180 min.

### 2.4. Activity Evaluation of Peak 13 from Pool II after Dantrole Treatment and of Repurified Peaks (28/29, 30, and 31) from Pool III of Phoneutria nigriventer on the Rat Body Temperature

To better evaluate the activity of the peaks that showed hyperthermic activity of pool II of *Phoneutria nigriventer* venom (8 and 13), an in vivo test was carried out together with the drug Dantrolene. For this in vivo test, peak 13 was determined due to its yield. Therefore, Dantrolene and peak 13 from pool II of *Phoneutria nigriventer* were tested simultaneously, both together (in the same animal) and separately in relation to interference with thermoregulation in rats. Figure 5 shows that peak 13 did not show significant activity in relation to controls during the three hours of the experiment. Animals treated with peak 13 and Dantrolene and animals treated only with Dantrolene also did not show significant activity in relation to the controls.

Considering that the peaks tested from pool III (Figure 5) showed thermal activity in rats (peaks 28/29, with hypothermic activity, and subpeaks 30 and 31, with hyperthermic activity), these materials were repurified by reverse phase chromatography (Appendix A). The chromatographic profile of peak 28/29 showed the presence of an asymmetric peak, indicating the possible presence of more than one molecule. In peaks 30 and 31, a non-symmetrical peak was observed, indicating the presence of other molecules. The peaks resulting from this purification were collected and tested for a better understanding of thermoregulation activities.

After repurifying the peaks from pool III of *Phoneutria nigriventer* venom that showed activity in thermoregulation (28/29, 30, and 31), in vivo tests were carried out to observe activity in relation to thermoregulation. The animals were administered with the peaks repurified; however, there was no significant change, as can be seen in Figure 5.

To better assess the activity of the peaks that showed hyperthermic activity in pool II of *Phoneutria nigriventer* venom (8 and 13), an in vivo test was carried out together with the drug Dantrolene. For this in vivo test, peak 13 was determined due to its yield. However, the animals treated with peak 13 and Dantrolene and the animals treated with Dantrolene alone showed no significant activity compared to the controls.

### 2.5. Evaluation of the Variation in Body Temperature Using the E. coli LPS

In this step, the peaks repurified from peak 28/29 were tested since peak 28/29 showed hypothermic activity. However, when these peaks were tested, it was only possible to observe that there was no significant activity in relation to the control. 

### 2.6. Identification by Mass Spectrometry of Peaks (8 and 13) with Action on Thermoregulation

Peaks 8 and 13 of pool II of *Phoneutria nigriventer* were analyzed by a mass spectrometer, and the masses identified in these peaks (Table 1 and Table 2) were compared with masses already described and recorded in the protein database, UniProtKB/Swiss-Prot, referring to proteins present in the arthropod venoms. In this identification, it was possible to detect, at both peaks, the presence of proteins from the Tachykinin family and the Ctenotoxin family, mainly of the Pn1a type.

## 3. Discussion

Among venomous species involved in accidents with humans in Brazil, *Phoneutria nigriventer* and *Tityus serrulatus* are responsible for most of the accidents. The clinical manifestations of accidents caused by *Phoneutria nigriventer* indicate central and peripheral actions [6]. Among several manifestations are pain, vomiting, tachycardia, hypotension, hypertension, arrhythmia, sweating, and tremors [4,6,7]. Arthropod venoms are a rich source of compounds and small active molecules with effects that remain undiscovered [8]. *Phoneutria nigriventer* venom is composed of several components [9] that act on channels of neuronal importance, such as sodium, calcium, and potassium channels. Gomez and collaborators (2002) quantified seventeen active peptides with functions involving ion channels and receptors [6] with weights between 3500 and 9000 Da; symptoms also include tremors and sweating, which are symptoms linked to thermoregulation. The venom of *Phoneutria nigriventer*, also acts on channels of neuronal importance, in this case, voltage-dependent sodium and potassium channels [10].

Therefore, it is possible to see that the symptoms described after accidents with *Phoneutria nigriventer* may also have different symptoms, and some researchers may describe them as antagonistic. These symptoms include hypotension and hypertension or hypothermia and hyperthermia [4,6,11]. Therefore, this work aims to evaluate the activity of the total venom of *Phoneutria nigriventer* and their isolated fractions in relation to variations in the body temperature of rats.

The aforementioned antagonistic effects can also be seen in tests carried out with animals treated with crude venom and purified fractions. Animals treated with the crude venom showed both hypothermic and hyperthermic action. Data obtained using the crude venom of *Phoneutria nigriventer* [12] showed a mainly hypothermic action in rats. The fractionation of the venom resulted in fractions containing molecules of different molecular weights, as expected, as it is a molecular exclusion chromatography, where high molecular weight molecules are eluted first, followed by medium [4,6,11] molecular weight molecules and the last molecules to be eluted are those of low molecular weight. Thus, pool I of *Phoneutria nigriventer* venom showed bands between 80 and 40 kDa and some less evident bands below 40 kDa. Pool II showed molecules with a molecular weight below 30 kDa, and pools III and IV showed molecules with a molecular weight between 30 and 20 kDa.

In order to study the obtained fractions activities on thermoregulation, the pools were tested in vivo, and it was possible to observe that pool II of *Phoneutria nigriventer* venom caused a decrease in body temperature in rats from the second hour of the experiment. However, pool III showed hyperthermic activity from the second hour onwards, but this did not maintain the increase in body temperature as pool II continued to maintain hypothermic activity. Therefore, the activity of pool III was more easily controlled by the rat organism than pool II. Studies show that accidents with *Phoneutria nigriventer* can cause, in some cases, tremors and/or sweating, highlighting the antagonistic effect of this venom [4,6,7]. This corroborates the data obtained in this work, where pool III caused hyperthermia and pool II hypothermia.

Therefore, in order to identify the possible components responsible for the activity on thermoregulation, the active pools were purified and tested in vivo. In this purification stage, it was possible to observe that pool II of *Phoneutria nigriventer*, with hypothermic action, presented a chromatogram containing 32 peaks. However, 6 of these 32 were chosen for in vivo tests, due to the peak yield, but none of these chosen was responsible for the hypothermic activity, presented by pool II. Although none of the chosen peaks showed hypothermic activity, peaks 8 and 13 were responsible for a significant increase in the rats’ body temperature in the third hour of the experiment in relation to control animals. Among the peaks chosen, peak 8 was the one with the lowest intensity, so it was only possible to attempt identification by mass spectrometry. Peak 13 was subjected to cell viability assessment using the MTT colorimetric method using human glioblastoma cells (U87) (Appendix A), as it is described that the venoms of these arthropods have many neurotoxins, that is, they have action on the cells of the nervous system [11]. Sueur and collaborators [13] also tested *Phoneutria nigriventer* venom in cell culture for viability, and these tests were carried out on three different cell types of neuronal origin: ECV304 endothelial, C6 glioma, and epithelial MDCK, showing that there was a small decrease in the viability of cells of the C6 glioma lineage, without interference in the other lineages [13].

Thus, as peak 13 is responsible for increasing body temperature, it was also tested together with the medication for malignant hyperthermia, Dantrolene. Malignant hyperthermia is a genetic disease in which the patient’s body temperature increases when subjected to some types of inhalational anesthetics such as halothane, isoflurane, or succinylcholine [14]. It is known that Dantrolene acts by blocking RYR1 and RYR3 type calcium channels, present in skeletal muscle and other tissues, including the brain, respectively [15]. Since *Phoneutria nigriventer* neurotoxins act on channels of neurological importance and also on the release of neurotransmitters in synaptic clefts [5,16], the synergism of this medicine with this peak isolated from *Phoneutria nigriventer* venom could assist in the study of this molecule. However, tests regarding the joint action of peak 13 with Dantrolene did not show significant results. Being a medicine for malignant hyperthermia and also a muscle relaxant, the expected result was that the temperature would drop in the group of animals treated only with the medicine; however, this was not the case. It is known that Dantrolene acts on specific calcium channels, receptors other than ryanodine, and these types of channels may have different locations and concentrations in humans and rats [15]. This could be a plausible explanation for these non-significant results, as well as the fact that the disease malignant hyperthermia is related to several genetic mutations, and the majority are associated with chromosome 19, the gene for the ryanodine receptor RYR1 [17,18]. As animals do not have the disease malignant hyperthermia, they may simply not have the ryanodine receptor modified to interact with the active ingredient of Dantrolene, which could be an explanation for the absence of effects in normal rats.

Tachykininins have widespread pharmacological actions, acting on the central nervous, cardiovascular, genitourinary, respiratory and gastrointestinal systems. Peptides from this group were found mammalian tissue and also in amphibian gut and insects and the main peptides were Substance P, neurokinin A and B having three identified re-ceptors NK1, NK2 and NK3 [19,20]. Substance P and related tachykinins have significant actions on the central nervous system particularly in brain injury [21]. Compounds of the Ctenotoxin family are described as the ctenotoxin U13-CNTX-Pn1a acting, according to the Swiss/Prot database, probably also as a neurotoxin and U19-CNTX-Pn1a is non toxic to mice and insects [22]. In order to identify which molecule was present in peak 13, it was necessary to analyze these peaks by mass spectrometry. As a result of this analysis, masses similar to molecules already described were obtained from the Tachykinin and Ctenotoxin families, mainly of the Pn1a type. These molecules are being studied and it is known that tachykinins have widespread pharmacological actions, acting on the central nervous, cardiovascular, genitourinary, respiratory and gastrointestinal systems. The compounds of the Ctenotoxin family are described as the ctenotoxin U13-CNTX-Pn1a acting, according to the Swiss/Prot database, probably as a neurotoxin and U19-CNTX-Pn1a is not toxic to mice and insects [22]. Both are neurotoxins produced by the species *Phoneutria nigriventer* [1,23,24]. 

Just as pool II of *Phoneutria nigriventer* venom was responsible for hypothermic activity, pool III also demonstrated hyperthermic activity. So, now analyzing the results for pool III, it can be seen that its peaks 30 and 31 increase the temperature in animals, while its peak 28/29 decreases body temperature in rats. Pool III of *Phoneutria nigriventer* spider venom showed hyperthermic activity; however, one of the peaks from this pool tested in bioassay decreased the temperature in rats. Once again, the results show that the venom has molecules with antagonistic actions [6]. These observed effects could be mediated by a pro-inflammatory action in the central nervous system [25,26,27].

In this way, the peaks from tested pool III and that showed activity on body tem-perature in rats were repurified to proceed to the next step, which involved the use of LPS from E. coli bacteria. The three peaks 28/29, 30, and 31, were not very pure, which is why the peaks were repurified. After separation, these subpeaks were tested again to find out which activity(ies) were concentrated. Subpeak 28/29, with hypothermic action, generated three more peaks, and peaks 30 and 31 generated two peaks each (29a, 29b, 29c, 30a, 30b, 31a, and 31b). Of the three peaks belonging to peak 28/29, the one (s) that showed hypo-thermic activity would be tested together with LPS to verify whether one mechanism of action would interfere with the other; however, none of the three peaks showed the expected activity. The venom has many molecules, and these can interact with each other to have a satisfactory effect on the state of poisoning. Scorpion venom is a good example, with alpha and beta neurotoxins acting together [16].

Of these seven repurified peaks tested in vivo, only peak 30b, due to yield, was subjected to amino acid sequencing by Edman degradation. The remaining peaks will be sequenced in future work. Peak 30b, despite not having shown activity on the body temperature of animals, was previously selected to be sequenced, considering the neurotoxin Tx2-6. This neurotoxin acts mainly by decreasing the inactivation of sodium channels [28,29,30] and can also cause, in rats, piloerection and tremors [30], which are responses controlled by thermoregulation [31].

Interestingly, Leite and collaborators (2012) also carried out tests with the brain, lung, and heart tissue of animals treated with this neurotoxin and described that the brain tissue was the least affected [29]. The lungs showed vascular congestion and alveolar hemorrhage, and the heart showed hemorrhage in the sub-endothelial tissue. The peaks 29, 30, and 31 were tested in cultured hypothalamus neurobasal cells; all the peaks showed a statistically significant decrease in cell viability at lower concentrations after 24 h of incubation. At 6 h of incubation, the statistics showed that the peaks 30 and 31 increased the viability of cells in higher concentrations. Before repurifying peaks 28/29, 30 and 31, cell viability tests were performed in neuro-basal cell culture. All peaks decreased cell viability of these cultures at low concentrations after 24 h of incubation (Appendix A). The animals treated by Leite et al. [29] were injected with a concentration 8× higher than those treated in the present work. Our data also showed that after the 6-h incubation period, peaks 30 and 31 at higher concentrations were positively significant for cell culture (Appendix A) [32].

## 4. Conclusions

Therefore, the changes observed with pools II and III of *Phoneutria nigriventer* demonstrated a very significant effect. Peaks 8 and 13 of pool II of *Phoneutria nigriventer* also showed an important action on thermoregulation. Preliminary data on these peptides show the similarity of proteins with similar masses already described, which correlate them with neurotoxic actions. The peaks of pool III of *Phoneutria nigriventer* venom also showed interesting data compared to the actions, probably involving synergism and without greatly affecting the viability of some cells of nervous origin. These effects appear to be mediated by a direct action on CNS receptors, taking into account the time of onset of effects. However, we cannot rule out the involvement of inflammatory mediators in the action, even though the venom has little inflammatory effect at the site of action. Another hypothesis would be action on the S.N.C. through other mediators that would trigger the effect on thermoregulation. One problem encountered was the absence of a pure pharmacological antagonist that would act on thermoregulation, and that was not related to antipyretic/anti-inflammatory mechanisms. Rats are resistant to changes in temperature, particularly mediated by inflammatory pathways; thus, variations in body temperature ranging only 1 degree could be considered quite significant when compared to other species like rabbits or guinea pigs commonly used for pyrogen detection. Molecules isolated from the venom may act for summation or synergistically, and other peptides present in the venom can have the opposite effect. The study of animal toxins is a source of new biomolecules and drug development. There is little data about toxins that act on thermoregulation, and the search for new compounds able to regulate thermic body setup by different pathways of the inflammatory thermic regulation could be very promising in this work ctenotoxin U13-CNTX-Pn1a showed a significant action on rat temperature and other toxins also displayed effects that seem to be in a different pathway of Dantrolene that could be a promising action to help in thermoregulation dysfunctions.

## 5. Materials and Methods

### 5.1. Animals

The project was approved by the Committee on Ethics in the Use of Animals of Instituto Butantan (CEUAIB) under protocol number 869/11.

In the present study, male Wistar–Han heterogenic albino rats (*Rattus norvegicus*) of conventional controlled sanitary status, weighing approximately 180 g, were obtained from the Central Biotério of the Butantan Institute and maintained at the Laboratory of Biochemistry and Biophysics Laboratory. These animals were housed in ventilated shelves in groups of 4 to 6 rats per polypropylene cage (49 × 34 × 16 mm) with a carved bed of *Pinnus* sp., Autoclaved. Within 2 or 3 days, the animals were used for the experimentation described later in this study. The animals were fed commercially specific feed (Nuvilab CR1-Nuvital), autoclaved, and filtered water; both supplied “ad libitum”, and kept in an environment with a photoperiod of 12/12 h light/darkness controlled by L & D timer.

### 5.2. Venom

Venom from spiders (*Phoneutria nigriventer*) were supplied by the Venom Section of the Scientific Development Division of the Butantan Institute in lyophilized form, ranging from 50 to 60 mg per batch, and were stored in a freezer (−20 °C).

### 5.3. Gel Filtration of Venom from Phoneutria nigriventer

The venom was diluted in a 2% acetic acid elution solution. After dilution, the venom was centrifuged for 30 min at 955 g in a MIKRO 200R Hettich Zentrifugen centrifuge. The supernatant was removed, and the precipitate was discarded. This step was repeated for another two or three times, totaling a volume of 1.5 mL at the end. The venom was applied to a glass column (74 × 2 cm) filled with Sephadex G-50M resin (Sigma, St. Louis, MO, USA). The run flow was maintained at 200 μL/min. The fractions were collected with the aid of a model collector type 2110 fraction collector Bio-Rad^®^ (South Granville, Australia), every 1.5 mL. After the collection, all the tubes were read in a spectrophotometer, Pharmacia Biotech, Ultrospec 2000, UV/visible spectrophotometer, with a wavelength of 280 nm. The concentrations of the samples were calculated through the refractive indices. All fractions collected were grouped according to the graph obtained by the optical density reading. The samples were lyophilized and stored in a freezer at −20 °C.

### 5.4. High-Performance Liquid Chromatography (HPLC) of the Active Fractions of Phoneutria nigriventer Venom

According to in vivo assays, the fractions (pools) that showed activity were purified by high-performance liquid chromatography (HPLC-Shimadzu 10-A, ClassVP software 5.0), using a C18 reverse phase column (LiChroCart 250-4 µm 250 mm × 46 mm), monitored by a variable UV detector set at 214 nm. The following solutions were used for elution: (A) 0.1% TFA (trifluoroacetic acid) (MERCK, Darmstadt, Germany) in ultrapure water; (B) 10% mobile phase A in acetonitrile.

The *Phoneutria nigriventer* venom pools obtained through gel filtration were purified in HPLC using a gradient from 5 to 100% of solution B in 45 min, at a flow rate of 1 mL per minute with the elution solutions (mobile phase) already described peaks were collected, lyophilized and stored at −20 °C.

### 5.5. Evaluation of the Activity of Phoneutria nigriventer Venom on Rat Body Temperature Variation

#### 5.5.1. Effects of Crude Venom

To evaluate the effect of total venom on body temperature variation, we used six animals treated with 600 μg/kg (IP) of crude venom from *Phoneutria nigriventer*. A control group received saline (0.2 mL, IP) under identical conditions. Animals were anesthetized with ketamine hydrochloride (25 mg/kg) and xylazine hydrochloride (10 mg/kg) intraperitoneally. A temperature sensor (ML 309 Thermistor Pod-°C Scale-AdInstruments^®^) was inserted subcutaneously into the back of the neck and connected to recording software (Lab Chart Powerlab-Windows 8.1.30). After 30 min of anesthesia, an intraperitoneal sub-lethal dose of the venom was administered. Controls were conducted seasonally with three or four animals every 1.5 months to account for annual basal temperature variations in normothermic animals. Body temperature was monitored every 5 min for 3 h.

#### 5.5.2. Effects of Venom Pools

After obtaining the venom pools of *Phoneutria nigriventer*, five groups containing five to nine animals each were established. The treated groups received 600 μg/kg (IP) of the venom pools, while the control groups received saline (0.2 mL, IP). The surgical procedure and the equipment used were the same as described previously.

#### 5.5.3. Effects of HPLC Venom Peaks

After identifying the pools that showed activity on body temperature variation, these were purified by chromatography (HPLC). The resulting peaks were collected and tested for activity. Six groups containing three to six animals each were established for peaks from one pool, with a control group of four animals. For the second pool, five groups of six to eight animals each were established, with a control group of five animals. All animals were treated with 100 μg/kg (IP) of the peaks, while control groups received saline.

#### 5.5.4. Effects after Treatment with Dantrolene

To further evaluate the hyperthermic peak’s activity, the drug Dantrolene (Sigma Aldrich) was tested in conjunction with this peak. Using the same method described previously, Dantrolene (10 mg/kg, IP) was first tested in a group of five animals. Subsequently, a group of three animals was treated with both Dantrolene (10 mg/kg) and the hyperthermic peak from *Phoneutria nigriventer* venom. Both treatments were administered 30 min after anesthesia.

#### 5.5.5. Effects after Repurification of Peaks from Pool III

After identifying the peaks from pool III that showed activity on body temperature variation, these were repurified by chromatography (HPLC). Seven groups were established for the repurified peaks 29a, 29b, 29c, 30a, 30b, 31a, and 31b, each containing three to five animals and a control group of five animals. All animals were treated with 100 μg/kg (IP) of the repurified peaks, while control groups received saline.

### 5.6. Identification by Mass Spectrometry of the Venom Peaks of Phoneutria nigriventer with Action on Thermoregulation

After identifying the peaks that showed some activity on the body temperature variation, they were submitted to identification using mass spectrometry by Dr. Daniel Carvalho Pimenta in the laboratory of Biochemistry and Biophysics of the Butantan Institute, in an ESI-I*T*- Tof (Shimadzu Co., Kyoto, Japan) spectrometer. The samples were diluted in 50% acetonitrile in water containing 0.5% formic acid and injected directly into the mass spectrometer by hand injection in a Rheodyne injector, positive mode, with a flow of 50 μL/min, in the same solution used in the dilution of the samples. The voltage of the interface used was 4.5 kV, and the voltage of the detector was 1.76 kV, with a temperature of 200 °C. The fragmentation was by argon collision gas, with 50% of energy, and the spectra were obtained in the range of 50 to 200 m/z. The data obtained were analyzed by LC-MS solution software 2.7.0 (Shimadzu Co.,Kyoto, Japan) and compared with the UniProtKB/Swiss-Prot database of venom proteins from *Phoneutria nigriventer* for identification.

### 5.7. Peptide Sequencing by Edman Degradation

For identification of pool III peak 30b, it was subjected to Edman degradation sequencing (Edman, 1950), as follows: 15 μL of the sample were applied to a glass fiber membrane treated with trifluoroacetic acid (TFA) (Wako), and its N-terminal end was sequenced by Edman degradation in a Shimadzu PPSQ-21 automatic protein sequencer (Shimadzu Co., Kyoto, Japan) following the manufacturer’s instructions. The Edman degradation technique consists primarily of the coupling of the peptide with the phenylisothiocyanate (PITC) reagent with the addition of pyridine to alkaline conditions, in which PITC reacts with the N-amino terminal residue to form N-phenylthiocarbamoyl (PTC). Under acidic conditions with the addition of TFA, a molecular rearrangement takes place in which the first peptide bond is cleaved, and, thus, the formation of anilinothiazolinone (ATZ) and the remainder of the peptide without the first amino acid. Still in a dilute acid medium, ATZ is converted to a more stable form in the form of phenylthiohydantoin (PTH). The quantification and identification of the samples are performed by comparing them with the standard analyzed at the start of the sequencing.

### 5.8. Statistical Analysis

Data related to body temperature variation tests were statistically analyzed using GraphPad Prism5 software. The test used was a *T*-test, and the data were analyzed every fifteen minutes, comparing the control group with the experimental groups.

## Figures and Tables

**Figure 1 toxins-16-00398-f001:**
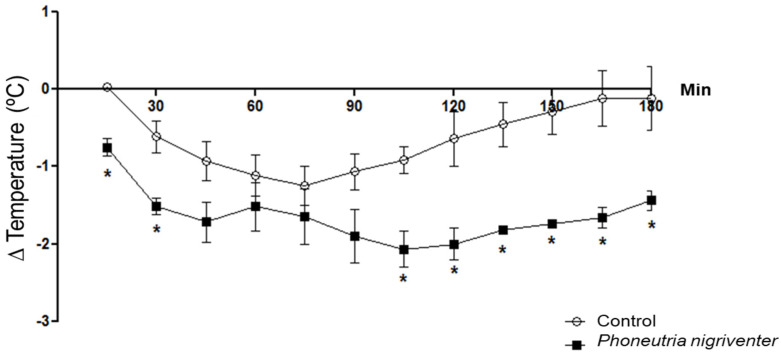
Variation in body temperature in rats after injection of crude *Phoneutria nigriventer* venom. The graph shows the average temperature variations of the control animals administered saline (IP) (n = 8) and the animals treated with 600 µg/kg (IP) of crude *Phoneutria nigriventer* venom (n = 6) and the temperature was measured every 15 min for 3 h. Significant values for *p* < 0.05 (*) using a *t*-test between the average temperature of the control animals and the experimental animals at each point (15, 30, 45, 60, 75, 90, 105, 120, 135, 150, 165, and 180 min).

**Figure 2 toxins-16-00398-f002:**
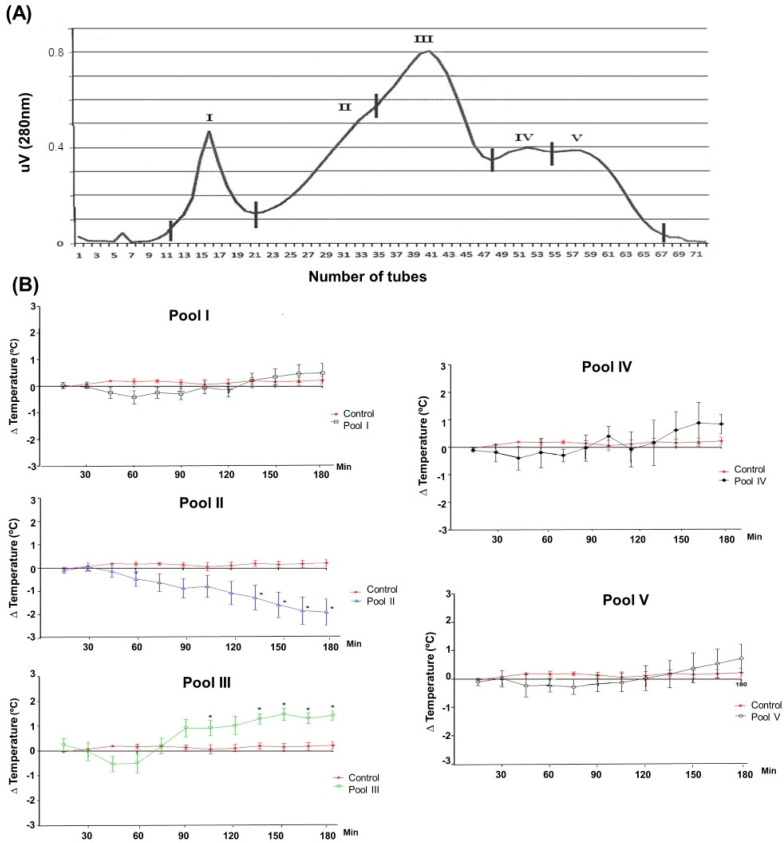
Fractionation of *Phoneutria nigriventer* venom (**A**) and body temperature variation in rats after injection of pools I, II, III, IV, and V (**B**). Body temperature variation in rats after injection of pools I, II, III, IV, and V of *Phoneutria nigriventer* venom, separated by gel filtration. Significant results are marked with an asterisk (*) on the graph curve. The average temperature variations of the control animals, administered intraperitoneal saline solution (n = 6), are compared with the animals administered intraperitoneally with 600 µg/kg of each pool of *Phoneutria nigriventer* venom, and the temperature was measured every 3 min for 3 h of the experiment. The *t*-test was performed between the average temperature of the control animals and each group of experimental animals at each time point (15, 30, 45, 60, 75, 90, 105, 120, 135, 150, 165, and 180 min).

**Figure 3 toxins-16-00398-f003:**
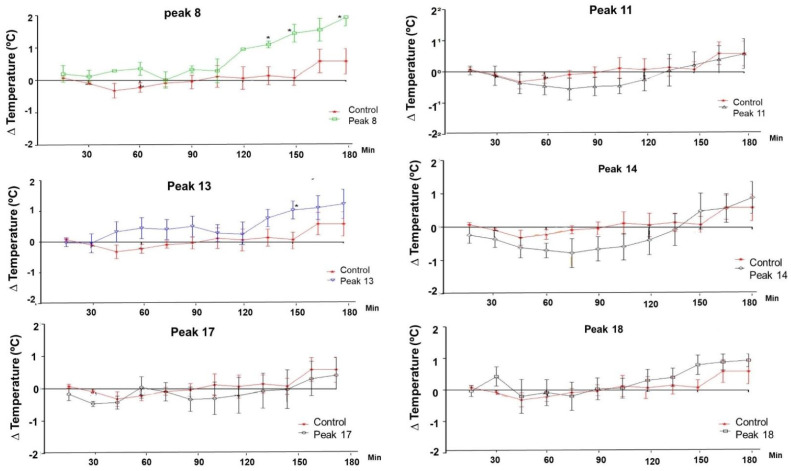
Body temperature variation in rats after injection of peaks 8 (n = 3), 11 (n = 5), 13 (n = 5), 14 (n = 6), 17 (n = 6), and 18 (n = 5) of pool II of *Phoneutria nigriventer*. Significant results are marked with an asterisk (*) on the graph curve. The average temperature variations of the control animals, administered intraperitoneal saline solution (n = 4), are compared with the animals administered intraperitoneally with 100 µg/kg of each material, and the temperature was measured every 15 min during the 3 h of the experiment. Significant values for *p* < 0.05 (*) were obtained by the *t*-test between the average temperature of the control animals and each group of experimental animals at each time point (15, 30, 45, 60, 75, 90, 105, 120, 135, 150, 165, and 180 min).

**Figure 4 toxins-16-00398-f004:**
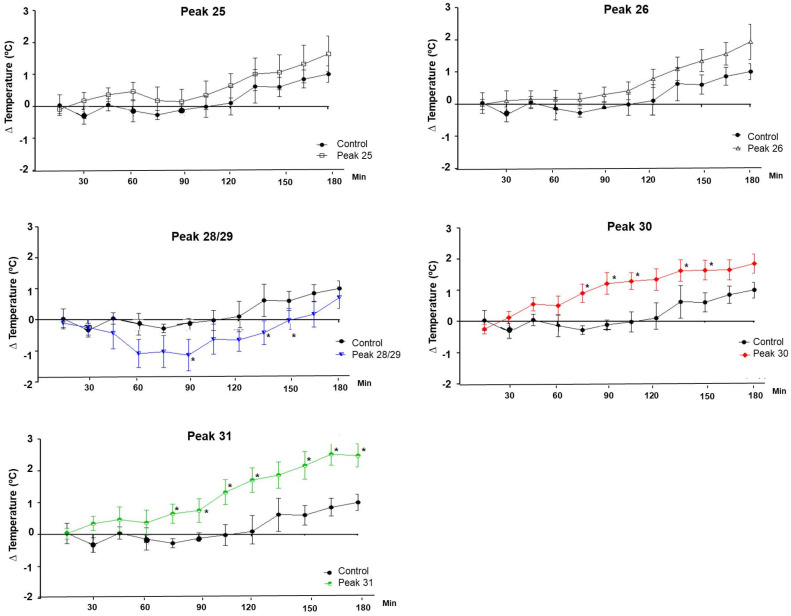
Body temperature variation in rats after injection of peaks 25 (n = 8), 26 (n = 6), 28/29 (n = 5), 30 (n = 6), and 31 (n = 6) from pool III of *Phoneutria nigriventer*. Significant results are marked with an asterisk (*) on the graph curve. Average temperature variations of the control animals, administered intraperitoneal saline solution (n = 6), are compared with the animals administered intraperitoneally with 100 µg/kg of material, and the temperature was measured every 15 min during the 3 h of the experiment. The *t*-test was performed between the average temperature of the control animals and each group of experimental animals at each time point (15, 30, 45, 60, 75, 90, 105, 120, 135, 150, 165, and 180 min).

**Figure 5 toxins-16-00398-f005:**
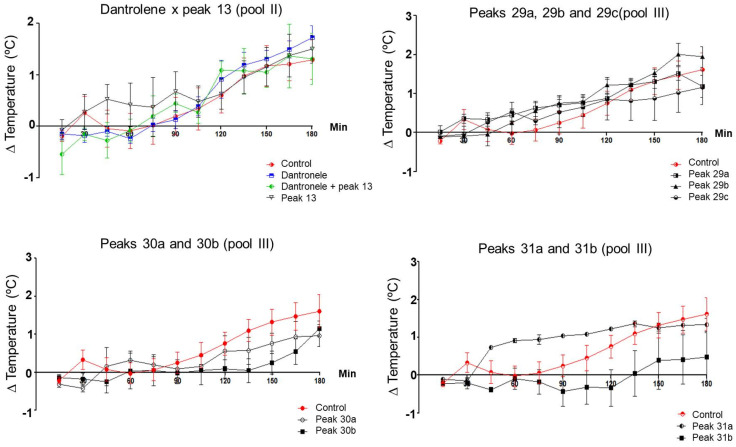
The graph showing the average temperature variations of the control animals, administered intraperitoneal saline solution (n = 4), compared to the animals administered intraperitoneally with 10 µg/kg of subpeaks 13 (n = 4) from pool II and 29a (n = 5), 29b (n = 5), 29c (n = 4), 30a (n = 3), 30b (n = 4), 31a (n = 3), and 31b (n = 4) from pool III of *Phoneutria nigriventer*. Significant results are marked with an asterisk (*) on the graph curve. Subpeak 13 was treated with 10mg/kg of the drug Dantrolene (n = 5). The *Phoneutria nigriventer* subpeaks were administered at 10 µg/kg and the temperature was measured every 15 min during the 3 h of experimentation. The *t*-test was performed between the average temperature of the control animals and each group of experimental animals (peak 13 of pool II of *Phoneutria nigriventer*, Dantrolene, and peak 13 of pool II of *Phoneutria nigriventer* + Dantrolene) at each time point (15, 30, 45, 60, 75, 90, 105, 120, 135, 150, 165, and 180 min).

**Table 1 toxins-16-00398-t001:** Identification of compound masses found by mass spectrometry of the peak 8 of Pool II of the venom of *Phoneutria nigriventer*.

Mass Found (Da)	Researched Mass (Da)	Amino Acid Number	Sequence	Description	Protein Accession
1139.74	1146.86	9	QKKDKKDKF	Tachykinin-like peptide-IV	P86301
		32	VFCRSNGQQC TSDGQCCYGK CMTAFMGKIC MR	U13-CNTX-Pn1a	P83894

**Table 2 toxins-16-00398-t002:** Identification of compound masses found by mass spectrometry of the peak 13 of the Pool II of the venom of *Phoneutria nigriventer*.

Mass Found (Da)	Researched Mass (Da)	Amino Acid Number	Sequence	Description	Protein Accession
1216.67	1217.00	10	QKKDRFLGLM	Tachykinin-like peptide-VI	P86303
1637.8	1637.93	13	QKKDKKDRFY GLM	Tachykinin-like peptide-XIII	P86310
3565.4	3549.5	32	FCRSNGQQC TSDGQCCYGK CMTAFMGKIC MR	U13-CNTX-Pn1a	P83894
7700.07	7666.6	68	QWIPGQSCTN ADCGEGQCCT GGSYNRHCQS LSDDGKPCQR PNKYDEYKFG CPCKEGLMCQ VINYCQKK	U19-CNTX-Pn1a	P83997

## Data Availability

The original contributions presented in the study are included in the article/Appendix A, further inquiries can be directed to the corresponding author.

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
