# Peer review of "Thermoregulation Effects of Phoneutria nigriventer Isolated Toxins in Rats"

_toxins, 2024, doi:10.3390/toxins16090398_

Round 1
Reviewer 1 Report
Comments and Suggestions for Authors
Dear authors,
Thermoregulation effects of spider venom have remained relatively understudied, and the analysis presented by the authors is welcome and very useful. The authors studied thermoregulation by applying venom and venom fractions/components on rats. Methods and results are sufficient. The analysis was well conducted and discussed.
However, I would like to offer a few suggestions and remarks about the information and analysis.
Minor remarks
Lines 77-79: The authors state that “the crude venom of Phoneutria nigriventer caused a significant decrease in the body temperature variation of the rats, especially between 105 and 180 minutes”. A decrease can be observed in figure 1, but it can also be seen that the effect diminishes after 120 min (also in the control). A question out of curiosity: did the authors also observe a change in behaviour in the rats later than 120 min?
Tables 1 and 2: I recommend to show only the components with the best molecular mass match (highlighted in yellow; mass difference of 1 Da). I also suggest to add the UniProtKB access numbers of the related peptides.
In line 293, the authors state that “As animals do not have the disease malignant hyperthermia, they may simply not have the ryanodine receptor modified to interact with the active ingredient of Dantrolene that could be an explanation to the absence of effects in normal rats.” However, Dantrolene has been used in different animal models, including rat models, and perhaps it is associated with different regulation pathways (RyR signaling) or affinities. Could that be the case?
Discussion: the components that were identified as Tachykinin-like peptides are very interesting. I recommend to discuss the activity of these peptides and their results more in detail (components 1216.67 Da and 1637.93 Da).
Figure S1: I suggest to add the amount of venom that was used for the pools, or the estimated amount of protein in the fractions (pools). Also, check the name of the reagent: “AccuBlock TM Digital Dry Bath, Labnet, Labnet Internat International, Inc.”
Figure S3: the figure caption states “these materials were purified by reverse phase chromatography (supplementary figure 3).” But I believe the authors refer to figure S2.
The footnote of Figure S4 states “incubation period of 24 hours”, although the figure says 6h.
Author Response
Firstly the authors are indebted to the comments in the manuscript. The authors point that these new data on thermic regulation by components of the venom of Phoneutria nigriventer and also in the studies to be developed from other venoms will be a new feature to be addressed that could be show new compounds that acts with different or more effective action leading in the future to the possible development of new pharmaceuticals.
From the suggestions raised by the referee to improve the text were as follow:
Lines 77-79- The effect of Phoneutria nigriventer crude venom is the result of the presence of a large number of different toxins with different actions, and pharmacokinetic and pharmacodynamic actions. After purification of fractions by gel filtration chromatography and isolation of peaks by HPLC different actions were revealed. Concerning changes in the behavior of the rats no significant features were observed, only a certain grade of restlessness and piloerection.
Tables 1-2- As suggested the tables were modified showing only the identified peaks.
In line 293- The observation that RyR signaling regulation or other kinds of effects could be related to the observed effect is an interesting point to be addressed in further studies including effects on the central nervous system such as brown adipose tissue culture that is one specific structure related to thermoregulation.
Discussion:
The effects of tachykinins was addressed in discussion in lines 302-308
Figure S1- The use of protein amount is the most common technique used, particularly for protein rich venoms and a good correlation with the amount of material is obtained. With some venoms venoms there are some issues that must be considered such as those with main polypeptide composition such as spider or some scorpion venoms (with Tityus serrulatus protein amount reflects hyaluronidase content and not the neurotoxins present in the venom) these controversial results specially after purifications steps result that Bradford and BCA methods doesn’t match, and also 280 nm readings. In this regard we have a choice to use dry weight to follow the purification steps that resulted in more confident results.
Figure S3 The type mismatch was corrected to (supplementary material S2)
The footnote of Figure S4 was corrected to 6 hours.
Expecting to have fulfilled all points raised properly, best wishes and regards

Reviewer 2 Report
Comments and Suggestions for Authors
Dear authors,
After reading your article, I’m drawn to this interesting work, which found a series of toxins purified from the venom of Phoneutria nigriventer can affect the body temperature of rats. Some of my points and suggestions are as follows.
First of all, the paper is generally well structured that the background of research is fully explained, the purpose of study is clear and the results are adequately elaborated and discussed. Secondly, though the role of toxins from Phoneutria nigriventer in regulating body temperature of rats can be clearly demonstrated by the part of results, the functional verification experiments of these peptides isolated by HPLC should be added as well for this study. Moreover, the latter parts of discussion, conclusion and reference are completed and articulated with clarity, making this research more accessible.
In general, a number of attractive toxins from Phoneutria nigriventer are discovered to adjust body temperature in rats. What’s even more exciting is that few similar studies focus on toxins from venomous animals affecting body temperature and this research serves an inspiration for subsequent related studies. Furthermore, this article shed a new light of activities of toxins and may serve as candidate drugs for thermoregulation dysfunctions and bring new tools to understand mechanism of thermoregulation in mammals.
Best wishes!
Sincerely,
A Reviewer
Comments on the Quality of English LanguageIt's basically error-free.
Author Response
Firstly the authors are indebted for the kind comments for the manuscript. The authors point that these new data on thermic regulation by components of the venom of Phoneutria nigriventer and also in the studies to be developed from other venoms will be a new feature to be addressed that could be show new compounds that acts with different or more effective action leading in the future to the possible development of new pharmaceuticals.
Best wishes and regards

Reviewer 3 Report
Comments and Suggestions for Authors
The reviewed article presents an interesting aspect of the study of the properties of venom components. I admit that this is the first article I have read that focuses on the thermoregulatory properties of venoms. Unfortunately, the article raises more questions than it answers and does not solve any research problem; it is merely an observation of certain phenomena and an inconsistent one.
As, in my opinion, the article is not suitable for publication in Toxins I will present only my biggest objections here. On the language issues, I will only mention that in all the captions of Figures 1-5, the content indicates that the venom/fraction was administered to the animals every 15 minutes for 3 hours, which is inconsistent with the description of the methodology.
The first objection concerns the use of Dantrole. Since it does not work on rats (as proven in this article), what is the point and purpose of this experiment? I understand that a negative result is also a result, but in this particular case, it contributes nothing and explains nothing in the issue under analysis. It is also very puzzling why in this experiment peak 13 did not show the properties shown in Figure 3.
A similar allegation applies to the experiment with LPS, for which, by the way, there is no description of the methodology. Why write in the article that something was done when nothing came out of it and, most importantly, it does not provide any explanation for the phenomenon under study.
However, my biggest concern is the peptide identification results presented. First of all, the raw MS data were not included. It is not possible these days to identify proteins or peptides on a mass spectrometer without fragmentation. On what basis were these identifications made in the article? Just based on m/z? Tables 1 and 2 highlights in yellow quote "the masses with sequences and weights related to molecules found in the peak". What does this mean and what do the other records not marked in yellow do in these tables? The biggest objection - the differences in masses between the observed signals and the "identified" ones are huge. In Table 1, the first one marked in yellow differs by 7 Da and the second one by 13! In Table 2 it is even worse, because, for example, the third one marked differs from the "standard" one by 33 Da. These are differences that make it impossible to identify. Identification should be made based on fragmentation spectra and comparison with sequences deposited in databases. On the other hand, before making a proper identification attempt, please consider whether you are sure everything is right with these samples. My second objection to these identifications concerns the masses of the "identified" peptides. In Table 1 are placed data on the identification of peak 8 from pool II. As the authors themselves point out in the discussion and as can be seen on the gels in fraction II there are proteins with masses of about 30 kDa. In contrast, the authors "identified" there peptides with masses of 1.1 and 3.5 kDa. Where did they come from in this sample? As the authors themselves describe in the discussion, size exclusion chromatography separates molecules based on their size, so such small peptides should come out of the column last. The same problem applies to Table 2, where peak 13 from Pool II was identified and the largest peptide identified is 7.7 kDa. While it is clear on the gel that Pool II has a very wide mass range, I would insist that it does not go below 10 kDa. By the way, wonder if the gel filtration went correctly since Pool II has smaller proteins than Pool III and IV. The peaks, except for the first one, on the chromatogram are not separated at all, so I suggest using a longer column/different bed or slowing down the flow rate.
Other minor comments concern the methodology: there is no information on what chromatograph the SEC was done on. What does "The concentrations of the samples were calculated through the refractive indices" mean? How does one know what the coefficient is if they are mixtures? There is no procedure for SDS-PAGE and for staining the gels. Why don't the gels have a stacking gel?
Finally, two general comments: the subject of the study was venom, not poison (e.g., Line 472, 236, 237, 255), and in chromatography we have peaks, not spikes (e.g., 165, 167, 180).
Author Response
From the comments of the Referee 3:
Firstly the authors are indebted for the kind comments for the manuscript. The authors point that these new data on thermic regulation by components of the venom of Phoneutria nigriventer and also in the studies to be developed from other venoms will be a new feature to be addressed that could be show new compounds that acts with different or more effective action leading in the future to the possible development of new pharmaceuticals.
Concerning the observations raised:
First paragraph of observations: Indeed by mistake the legend of the figures was wrong suggesting that toxin and its fractions were administered every 15 minutes.
The legends were corrected to: treated with...and the temperature was measured each….
Second paragraph of observations: The use of Dantrolene was explained in the text of discussion since there are few compounds that act on thermic regulation with different mechanisms of the anti thermic compounds that act on inflammatory pathways. The possibility that ryanodine receptor(RyR) signaling regulation or other pathways for the observed effects could be related to the observed effect is an interesting point to be addressed in further studies including effects on the central nervous system such as brown adipose tissue culture that is one specific structure related to thermoregulation. The lack of effect observed by the use of Dantrolene gives us the information that the thermic effect of Phoneutria nigriventer (peak 13 pool II) has a different pathway from Dantrolene.
Concerning the effect of peak 13(pool II) in Figure 3 and 5 alone and with Dantrolene we does not see any discrepancy since in all experiments the increase of temperature was around 1 degree, some variations are intrinsic for “in vivo” experiments
Third paragraph of observation: The use of LPS follows the same rationale of Dantrolene since LPS is a very well known hyperthermic compound that increases the temperature of mammals by inflammatory pathways. The lack of effect suggests that there are no inflammatory pathway components in the action of the isolated peak.
Fourth paragraph of observations: The present work has not the intention to fully characterize the compounds. The main objective was to study some effects that were initially only reported in clinical reports and to better verify these effects in animal models “in vivo” of the venom, its fractions and HPLC peaks concerning this effect . A first preliminary molecular identification approach was done only in order to identify in which class of molecules from Phoneutria nigriventer venom these toxins belong. Otherwise, this identification was given as described in the manuscript by mass spectrometry generated data of the peaks and also N-terminus sequencing by Edman as described in Methods, the obtained data when searching in the protein data bank, and the data taken together that gave us the confidence for the active isolated toxin identification. It is important to point out that identification is not obtained only from mass spectra data as suggested. The table was modified and now includes only the active identified toxins with the protein accession. In future studies have to be done specially when the mechanism will be studied in rats or specific cell culture model related to thermal regulation such as brown tissue but but in this first approach and we show a relevant effect on thermoregulation in rats (that are quite resistant to thermic alterations), these effects revealed to be not related to Dantrolene effects (Rianodine receptors) or LPS (inflammatory pathways).
Fifth paragraph of observations the gel filtration chromatography was performed as described in Method 5.3 previously described in a regular system and runned by gravity feeding of the solvent and monitored at 280 nm. The pool amount was estimated by the reading in 280 (each pool) and after lyophilization by the dry weight. SDS PAGE electrophoresis was performed as described by Laemli and the description included in the manuscript. Is important to point out that the stained gels only show some high molecular weight compounds of the venom and are not suitable for polypeptides for this reason is included in the supplementary material and only was performed at the initial step of purification. The obtained result of Phoneutria nigriventer purification(SEC and SDS-PAGE) is in accordance with many previously published works (https://doi.org/10.1016/0041-0101(91)90195-W; https://doi.org/10.1016/0041-0101(94)00130-Z; https://doi.org/10.1007/bf00969702
Last paragraph of observations- the word mismatches present in the lines 472, 236, 237, 255 (venom instead poison), 165, 167 and 180 peaks instead spikes were corrected
Expecting to have fulfilled all points raised properly, best wishes and regards.

Round 2
Reviewer 3 Report
Comments and Suggestions for Authors
Thank you to the authors for clarifying aspects that, in my opinion, were not clear from the content of the article. The revised tables with peptide identifications now look much better. The current version will definitely be more accessible to readers. I wish you good luck in exploring this new research topic.